# Identification and Quantification of Jaundice by Trans-Conjunctiva Optical Imaging Using a Human Brain-like Algorithm: A Cross-Sectional Study

**DOI:** 10.3390/diagnostics13101767

**Published:** 2023-05-17

**Authors:** Takuya Kihara, Takaaki Sugihara, Suguru Ikeda, Yukako Matsuki, Hiroki Koda, Takumi Onoyama, Tomoaki Takata, Takakazu Nagahara, Hajime Isomoto

**Affiliations:** Division of Gastroenterology and Nephrology, Department of Multidisciplinary Internal Medicine, Faculty of Medicine, Tottori University, Yonago 683-8504, Japan; t.kihara@tottori-u.ac.jp (T.K.); ikeda_suguru@tottori-u.ac.jp (S.I.); matsukiy@tottori-u.ac.jp (Y.M.); hkoda@tottori-u.ac.jp (H.K.); t-onoyama@tottori-u.ac.jp (T.O.); t.nagahara@tottori-u.ac.jp (T.N.); isomoto@tottori-u.ac.jp (H.I.)

**Keywords:** smartphone, image processing, jaundice, hyperbilirubinemia, conjunctiva, cirrhosis, biliary obstruction

## Abstract

Jaundice is caused by excess circulating bilirubin, known as hyperbilirubinemia. This symptom is sometimes caused by a critical hepatobiliary disorder, and is generally identified as yellowish sclera when bilirubin levels increase more than 3 mg/dL. It is difficult to identify jaundice accurately, especially via telemedicine. This study aimed to identify and quantify jaundice by trans-conjunctiva optical imaging. Patients with jaundice (total bilirubin ≥3 mg/dL) and normal control subjects (total bilirubin <3 mg/dL) were prospectively enrolled from June 2021 to July 2022. We took bilateral conjunctiva imaging with a built-in camera on a smartphone (1st generation iPhone SE) under normal white light conditions without any restrictions. We processed the images using an Algorithm Based on Human Brain (ABHB) (Zeta Bridge Corporation, Tokyo, Japan) and converted them into a hue degree of Hue Saturation Lightness (HSL) color space. A total of 26 patients with jaundice (9.57 ± 7.11 mg/dL) and 25 control subjects (0.77 ± 0.35 mg/dL) were enrolled in this study. The causes of jaundice among the 18 male and 8 female subjects (median age 61 yrs.) included hepatobiliary cancer (*n* = 10), chronic hepatitis or cirrhosis (*n* = 6), pancreatic cancer (*n* = 4), acute liver failure (*n* = 2), cholelithiasis or cholangitis (*n* = 2), acute pancreatitis (*n* = 1), and Gilbert’s syndrome (*n* = 1). The maximum hue degree (MHD) optimal cutoff to identify jaundice was 40.8 (sensitivity 81% and specificity 80%), and the AUROC was 0.842. The MHD was moderately correlated to total serum bilirubin (TSB) levels (*rS* = 0.528, *p* < 0.001). TSB level (≥5 mg/dL) can be estimated by the formula 21.1603 − 0.7371 × $\sqrt{{\left(56.3-\mathrm{MHD}\right)}^{2}}$. In conclusion, the ABHB-based MHD of conjunctiva imaging identified jaundice using an ordinary smartphone without any specific attachments and deep learning. This novel technology could be a helpful diagnostic tool in telemedicine or self-medication.

## 1. Introduction

The incidence of jaundice is approximately 40,000 per 100,000 intensive care unit patients [1]. Neonatal jaundice is more common among infants of Southeast and Far East Asian descent compared to those of Caucasian or African descent [2]. In adults, bile duct stones, cancers (pancreatic cancer, cholangiocarcinoma, and others), and liver disease attributed to excess alcohol are reportedly common causes of jaundice [3].

Jaundice can be a life-threatening condition for both neonates and adults. Identifying this condition is difficult even for experienced physicians. Physicians detect the presence or absence of jaundice with sensitivities and specificities of approximately 70% [4]. Their predictions varied from the measured value on average by 3.4 ± 5.3 mg/dL for serum bilirubin.

Commercially available transcutaneous bilirubinometers, such as the BiliCare System (Natus Medical Inc., Middleton, WI, USA) and JM-105 (Konica Minolta Inc., Osaka, Japan), employ light-based spectroscopy at different wavelengths to assess neonatal jaundice by measuring relevant substances, thereby determining hyperbilirubinemia. Transcutaneous bilirubinometers are specialized meters that non-invasively and indirectly measure bilirubin levels in neonates with jaundice. It is cost-effective and avoids unnecessary blood tests, but it tends to overestimate the total serum bilirubin (TSB) level at <12 mg/dL and underestimates it at a higher TSB level [5,6]. Transcutaneous bilirubinometers are priced in the range of USD 3000 to 5000. Consequently, the utilization of smartphones presents a cost-effective alternative for evaluating jaundice.

“BiliCam” is a smartphone application developed to detect the skin color of neonatal jaundice [7]. Their system needs two images, one with and one without the flash, and requires calibration to adjust skin colors.

Previous methodologies primarily concentrated on identifying jaundice in the skin of newborns; however, Leung et al. shifted their attention to the sclera and put forward the Jaundice Eye Color Index (JECI) as an alternative to assessing neonatal jaundice based on skin color [8]. In line with this approach, Outlaw et al. introduced a smartphone application named “neoSCB”, specifically designed for evaluating jaundice in neonates through eye examinations [9]. This application requires the acquisition of two sets of images: one with flash and the other without flash, focusing on the sclera. This application demonstrated a sensitivity of 100% and a specificity of 54% when screening infants with TSB above 12 mg/dL. Similarly, Mariakakis et al. developed “Biliscreen”, another smartphone application with the objective of assessing an individual’s bilirubin levels through the acquisition of eye images [10]. This application demonstrated an excellent sensitivity of 89.7% and a specificity of 96.8% in the detection of TSB ≥ 1.3 mg/dL related to liver and pancreatic disorders. Nevertheless, the application requires a specialized shading box.

Park et al. recently investigated a deep-learning system for detecting jaundice caused by hepatobiliary and pancreatic diseases using a smartphone [11]. This report also demonstrated excellent hyperbilirubinemia (TSB ≥ 1.5 mg/dL) prediction sensitivity (80.0%) and specificity (92.6%). However, their system also requires a color consistency patch.

Deep learning is commonly employed for image recognition and yields excellent accuracies [12,13]. However, a significant drawback of the technique is the requirement for extensive training data. To overcome such limitations, the Zeta Bridge Corporation (Tokyo, Japan; https://www.zeta-bridge.com/) developed a human brain-like algorithm, recognizing that these image features can provide critical insights. By leveraging this approach, the algorithm can detect visual characteristics that traditional deep learning algorithms cannot, hence improving image recognition performance. The algorithm is named Algorithm Based on Human Brain (ABHB). ABHB was initially designed for deployment in foreign object detection systems on production lines. The ABHB’s color analysis technology presents a novel approach to accurately obtain color information from images captured using ordinary cameras, regardless of the ambient light color conditions. Therefore, ABHB is independent of any light sources and does not require a color consistency patch. The present methodology enables an assessment to be conducted at any given time and location, obviating the necessity for specialized supplementary apparatus.

Here, we aimed to identify and quantify jaundice in the eyes of adult patients by ABHB using a smartphone camera.

## 2. Materials and Methods

### 2.1. Study Design and Protocols

This was a single-center, prospective, cross-sectional study. The inclusion criteria for this study were adult participants aged between 20 and 79 years, presenting with jaundice. The disease group comprised adults diagnosed with jaundice (total bilirubin ≥ 3 mg/dL according to the classical definition [14], while the control group must have been free of jaundice (total bilirubin < 3 mg/dL) upon blood testing. Participants exhibiting ocular conjunctival hyperemia or ophthalmologic conditions were excluded from the study. Additionally, the investigator reserved the right to exclude any participant deemed unsuitable for inclusion in the study. Between November 2021 and September 2022, we enrolled patients with jaundice and normal control subjects in this study. The sample size was between 24 and 50 according to the recommendation for feasibility and pilot studies [15,16]. Two patterns (straight and upturned eyes) of facial images were captured for each patient by a built-in camera equipped on a smartphone (1st generation iPhone SE, Apple Inc., Cupertino, CA, USA) under normal white light without flash from a 10 to 30 cm distance. The rationale behind selecting a smartphone camera as the imaging device instead of a specialized camera was rooted in the fact that approximately 5 billion individuals currently utilize mobile devices [17]. Therefore, this study intended to leverage the popularity of smartphone cameras and utilize them as medical imaging devices.

Each image was saved in JPEG format. Images were sent to Zeta Bridge Corporation, affiliated with Sony Corporation, and processed using an ABHB and converted into hue degree of Hue Saturation Lightness (HSL) color space (Figure 1). The ABHB business was recently transferred to ForgeVision, Inc. (Tokyo, Japan; https://www.forgevision.com/ accessed on 16 May 2023). Image recognition can be applied to issues that are not suitable for deep learning because ABHB does not require large-scale imaging data sets.

The main part of the compensation algorithm consists of background masking, noise reduction, and adjustment luminance, which the human brain naturally uses for recognition (Figure 2A,B). This algorithm minimizes the influence of circumstances such as the light source or distances. Finally, the color information is converted to an HSL image (Figure 2C). Bilateral eyes, including the surrounding small part of the skin, are automatically extracted from the two image patterns and the measured pixels of each hue degree (Figure 2D and Figure 3A). The highest maximum hue degree (MHD) was adopted for further analysis (Figure 3B and Figure 4). We obtained the patient’s data, such as biological gender, age, underlying diseases, and total serum bilirubin (TSB) levels.

### 2.2. Statistical Analysis

We applied Welch’s *t*-test to compare the two groups as defined by the cutoff criteria. We used the Spearman rank-order correlation coefficient (shown as *rS*) to evaluate the correlation between two variables, and Kolmogorov–Smirnov Test to analyze the distribution normality. All statistical tests were performed using StatFlex (Windows ver. 7.0; Artech, Osaka, Japan). Values are expressed as median (interquartile range) or mean with a standard deviation (SD). Categorical variables are shown as numbers. The statistical significance was set at *p* < 0.05.

## 3. Results

The baseline patient characteristics are presented in Table 1. A total of 51 patients (39 men and 12 women, median age 66 (51–72) years) were enrolled in this study. The study sample included 26 patients (18 men and 8 women, median age 61 (50–69) years) with jaundice and 25 control subjects (21 men and 4 women, median age 69 (51–73) years). The causes of jaundice included hepatobiliary cancer (*n* = 10), chronic hepatitis or cirrhosis (*n* = 6), pancreatic cancer (*n* = 4), acute liver failure (*n* = 2), cholelithiasis or cholangitis (*n* = 2), mass-forming pancreatitis (*n* = 1), and Gilbert’s syndrome (*n* = 1). The TSB of the jaundice group and the control subjects was 9.57 ± 7.11 and 0.77 ± 0.35 mg/dL, respectively.

The maximum hue degree (MHD) was obtained from images of straight eyes in 36 people and upturned eyes in 15 patients. The average MHD was 37.2 ± 6.6 in the normal group and 54.4 ± 16.1 in the subjects with jaundice. The optimal cutoff of the MHD to identify jaundice was 40.8 (sensitivity 81% and specificity 80%), and the AUROC was 0.842 (Figure 5A). The MHD was moderately correlated to total bilirubin levels (*rS* = 0.528, *p* < 0.001), and TSB was highest at an MHD of 56.3 (Figure 5B). The TSB level seemed higher when the MHD was close to around 56.3. The histogram of patients with jaundice demonstrated a normal distribution, and the median MHD was 54.8 (43.6–64.7) (Figure 5C). It indicates that the MHD of the highest TSB level is close to the median MHD. The range of jaundice distributes from an MHD of 40.8 to 100, and around an MHD of 56.3 is considered “typical” and “severe” jaundice.

It is suggested that the farther away from an MHD of 56.3 (the highest TSB), the lower the TSB value. Thus, the deviation from 56.3 was calculated by $\sqrt{{\left(56.3-\mathrm{MHD}\right)}^{2}}$. The TSB level and the deviations were moderately negatively correlated to a TSB > 5 mg/dL (*rS* = −0.662, *p* = 0.003) (Figure 5D). However, this formula is unsuitable for quantifying TSB 3–5 mg/dL because some cases of lower-range TSB are distributed around the MHD of 56.3.

## 4. Discussion

In this pilot study, we demonstrated that ABHB-based conjunctiva optical imaging could non-invasively identify and partially quantify jaundice. This is the first report on novel image recognition technology for identifying and quantifying jaundice in adults using a smartphone without any specific attachments and deep learning.

The study of objective color perception has a long history. In 1931, the Commission Internationale de l’Éclairage (CIE) proposed a color measurement system, which laid the foundation for contemporary colorimetry. This system enables the determination of color matches by employing CIE XYZ tristimulus values for color specification. The XYZ color space, also known as the CIE 1931 XYZ color space, is a mathematical representation of colors based on the trichromatic theory of human vision [18]. It exhibits “color differences”.

The “BiliCam” study used the XYZ color space [9]. The authors dedicated their efforts to implementing ambient subtraction by capturing two images: one with flash and another without flash. On the other hand, Padidar et al. used HSI color space with machine learning [19]. They used a color calibration card and 100X microscope on the built-in camera of the phone. They reported that the estimation of bilirubin levels had a correlation of 0.479 with the total serum bilirubin values. Our result (0.528) was better than theirs without any color calibration cards, special microscopes, or datasets for machine learning.

Among color spaces, HSL, HSI, and HSV are perceptually based color models that simplify the representation of colors by separating hue, saturation, and lightness (or intensity/value) components. In graphic applications, HSL and HSI are two alternative color spaces used to represent colors. The conversion between XYZ and HSL/HSI enables the translation and manipulation of colors between these color spaces, catering to various color-related applications and requirements. HSL and HSI color spaces are generally regarded as equivalent. The main difference between HSL and HSI is in the way they define the brightness or lightness of a color. Notably, the HSL color space primarily concentrates on the perceptual dimensions of color, with a particular emphasis on the human perception of hue [20].

Most other studies using smartphones maintained consistent conditions by fixing the light source and distance or employing a color consistency patch [7,8,9,10,11,19,21,22,23,24]. Typically, cameras modify color information via their built-in auto white balance function to minimize the effect of ambient light colors, ensuring that captured images appear natural when viewed by human observers. Due to variations in change logic across camera models and manufacturers, restoring accurate color information is currently a formidable task. The measuring instruments of color images generally give different results depending on the light source conditions. The ABHB system presents an innovative approach that obviates the requirement for both, enabling any device to perform objective judgments akin to the cognitive processes of the human brain, regardless of the prevailing light source.

Deep learning exhibits great promise as a technology with substantial potential for further advancement within the realm of medical applications. Kalbande et al. evaluated two models, ResNet50 and Detectron-2, for detecting jaundice [13]. They employed a pre-trained dataset that underwent fine-tuning specifically for jaundice detection, resulting in an accuracy of approximately 95%. Although their results were commendable, the manuscript lacks pertinent details concerning the background data, including the levels of jaundice. Undoubtedly, deep learning will serve as an influential tool for image recognition. However, ABHB presents an alternative approach that circumvents the necessity for extensive datasets.

We have previously reported on predicting risky esophageal varices by only taking an abdominal picture using a smartphone application [25]. We believe that in the future, the acquisition of physical parameters from images of patients will become a standard practice in clinical settings, particularly in telemedicine and self-medication.

Moreover, we envision that the integrated camera of an ABHB-equipped smartphone, with its potential for evolution, could transform into a visionary recognition system reminiscent of the healthcare-providing prototype robot “Baymax” portrayed in the Disney animated series.

This study has several limitations. First, the cohort studied represented a small group of patients with jaundice. Therefore, selection bias was inevitable. However, based on the promising results of this pilot study, a large-scale cohort study will be conducted for validation. Second, quantification was not well demonstrated between 3 and 5 mg/dL. It was because of the case variability in the lower range of TSB; a large-scale cohort study would also improve the correlation.

## 5. Conclusions

In conclusion, the implementation of ABHB-based luminance and color adjustments allows for the development of a straightforward color assessment system that is independent of the light source and distance. Our system does not rely on color consistency patches or deep learning techniques, yet it demonstrated the capability to detect total serum bilirubin levels of 3 mg/dL with sensitivities and specificities reaching approximately 80%.

## Figures and Tables

**Figure 1 diagnostics-13-01767-f001:**
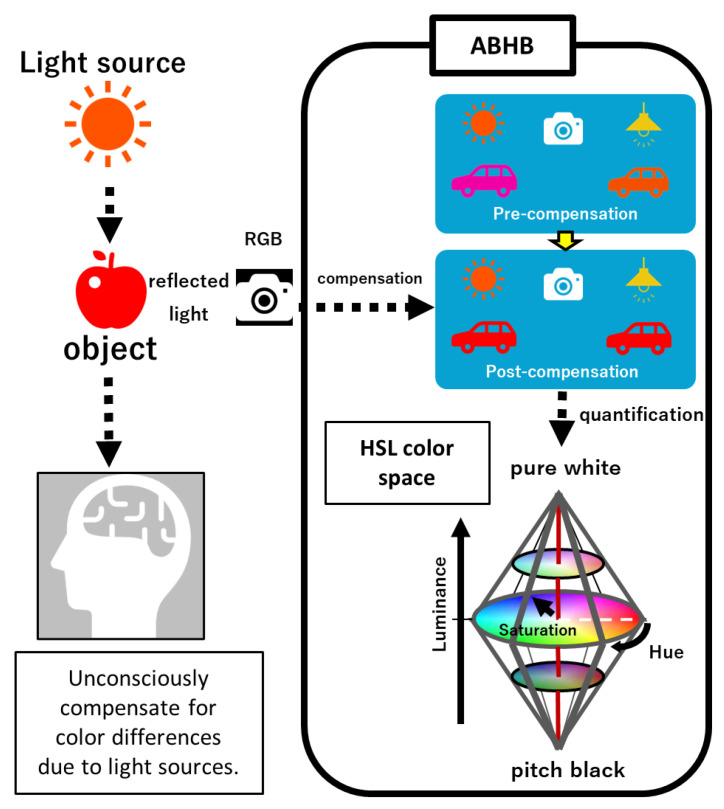
The ABHB concept. When humans see a subject under a light source, the brain unconsciously corrects the images. First, ABHB compensates for differences due to the light sources of the RGB images. Subsequently, the RGB color values are numerically converted to hue, saturation, and luminance values in the bi-conical 3D color space. ABHB, Algorithm Based on Human Brain; HSL, Hue Saturation Luminance; and RGB, Red Green Blue.

**Figure 2 diagnostics-13-01767-f002:**
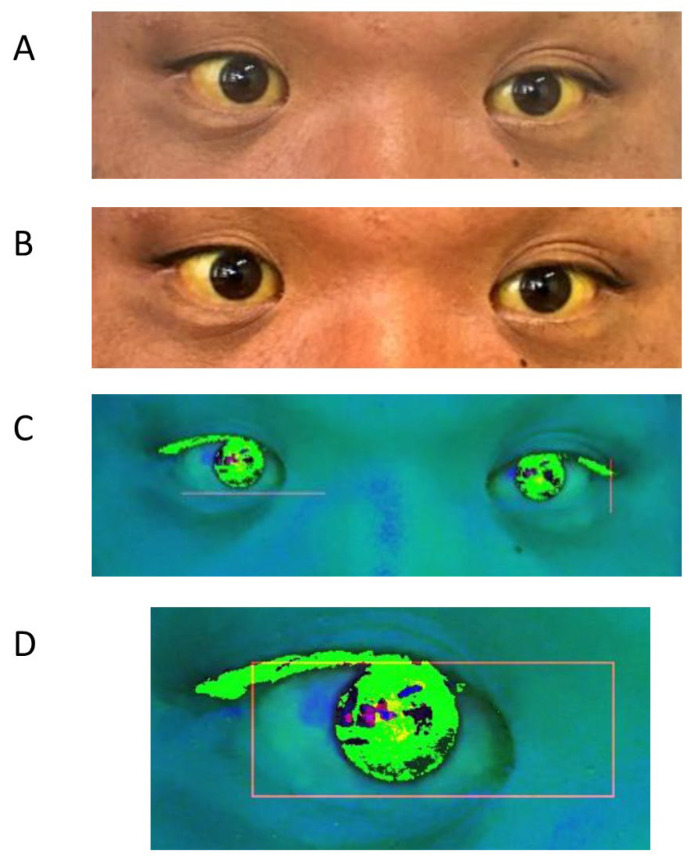
The ABHB process for capturing eye images. (**A**) The normal captured eye image. (**B**) The compensated image. (**C**) The HSL image. (**D**) Eye detection. ABHB, Algorithm Based on Human Brain; HSL, Hue Saturation Luminance.

**Figure 3 diagnostics-13-01767-f003:**
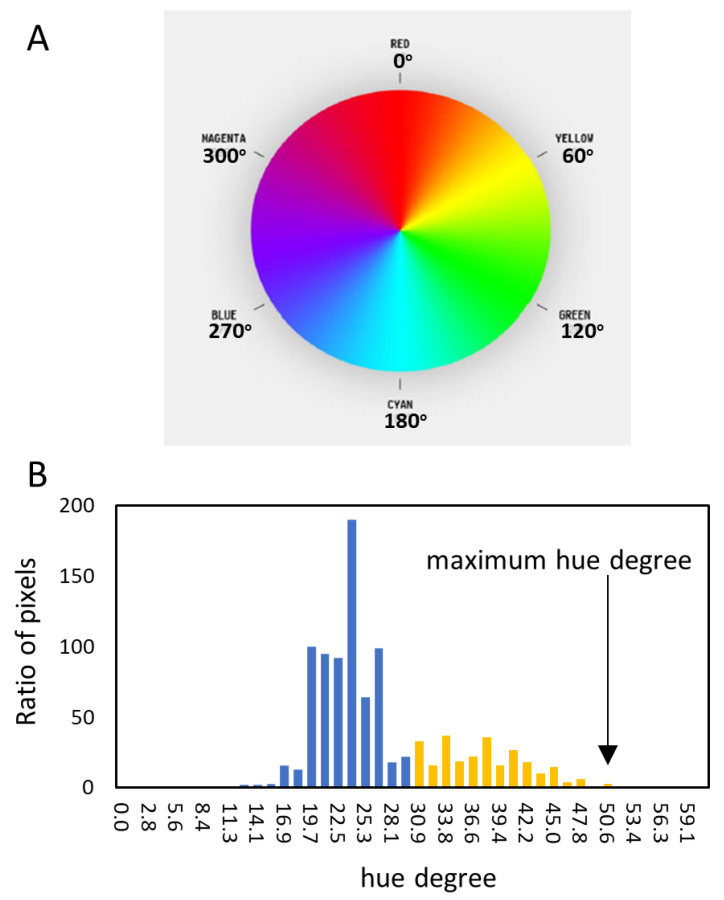
The hue degree color map and its histogram. (**A**) The color map of the hue degree. Yellow is around 60 degrees. (**B**) A representative histogram of hue degree. The highest value is the maximum hue degree (MHD) (arrow).

**Figure 4 diagnostics-13-01767-f004:**
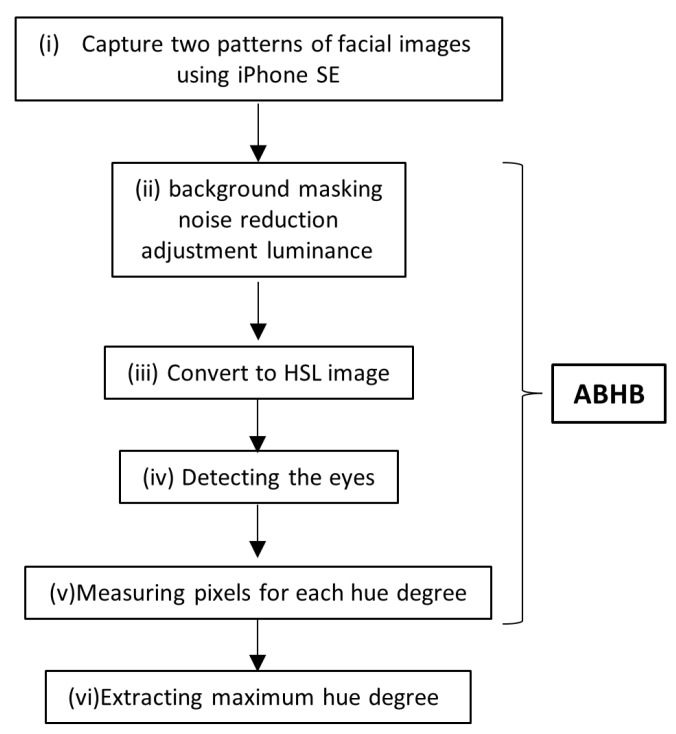
Study scheme. (i) Capturing the two patterns (straight and upturned eyes) of facial RGB images using an iPhone SE. (ii) Compensating via background masking, noise reduction, and adjustment luminance. (iii) Conversion to an HSL image. (iv) Detecting the eyes. (v) Measuring pixels for each hue degree. (vi) Extracting the maximum hue degree. The algorithm processing from (ii) to (iv) is called ABHB. ABHB, Algorithm Based on Human Brain; HSL, Hue Saturation Luminance; and RGB, Red Green Blue.

**Figure 5 diagnostics-13-01767-f005:**
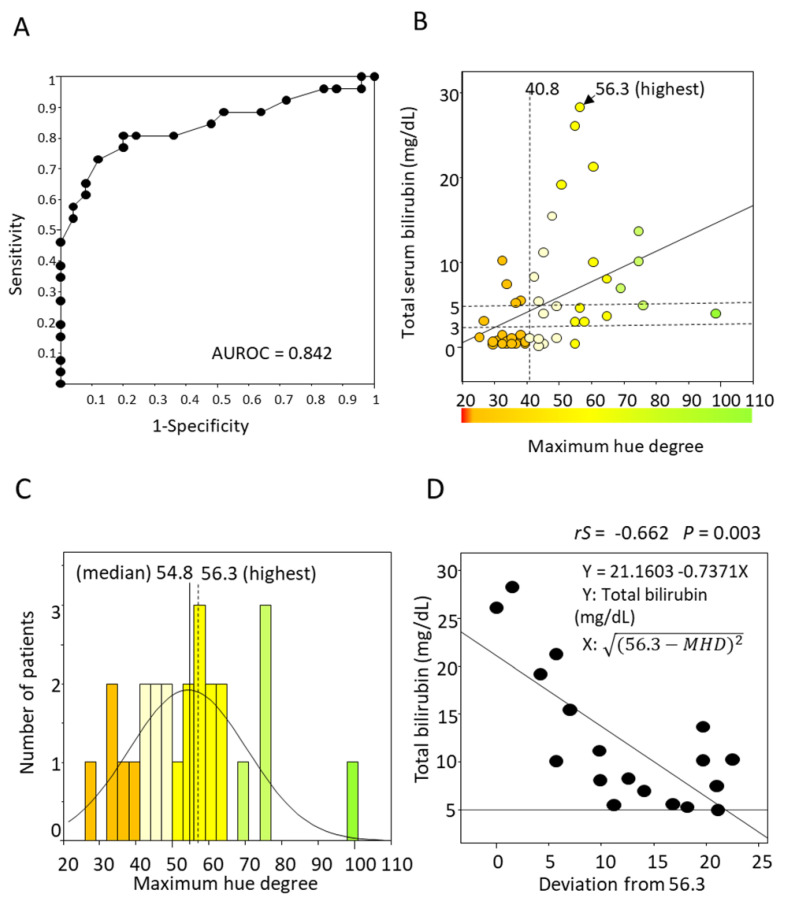
Receiver Operating Characteristic curve of MHD for detecting jaundice and the correlation of TSB and MHD. (**A**) The optimal MHD cutoff to identify jaundice was 40.8 (sensitivity 81% and specificity 80%), and the AUROC was 0.842. (**B**) MHD was moderately correlated to TSB levels (*rS* = 0.510, *p* < 0.001). TSB was highest at MHD = 56.3. (**C**) The histogram of jaundice cases had a normal distribution. The median MHD was 54.8, and the TSB was highest at MHD = 56.3. (**D**) TSB level (≥ 5 mg/dL) can be estimated by the formula 21.1603 − 0.7371 × $\sqrt{{\left(56.3-\mathrm{MHD}\right)}^{2}}$. MHD, maximum hue degree; TSB, total serum bilirubin.

**Table 1 diagnostics-13-01767-t001:** Patient characteristics.

Patients	Control (*n* = 25)	Jaundice (*n* = 26)
Sex (Male/Female)	21: 4	18: 8
Median age (years)	69 (51–73)	61 (50–69)
Total bilirubin (mg/dL)	0.77 ± 0.35	9.57 ± 7.11
Underlying diseases	chronic hepatitis or cirrhosis (9) liver cancer (3) acute pancreatitis (1) cholangitis (1) others * (11)	hepatobiliary cancer (10) chronic hepatitis or cirrhosis (6) pancreatic cancer (4) acute liver failure (2) cholelithiasis or cholangitis (2) mass-forming pancreatitis (1) Gilbert’s syndrome (1)

***** Others included patients with colon polyps (4), early gastric cancer (2), ulcerative colitis (1), pyogenic spondylitis (1), and healthy volunteers (3).

## Data Availability

Not applicable.

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
