# Peer review of "Identification and Quantification of Jaundice by Trans-Conjunctiva Optical Imaging Using a Human Brain-like Algorithm: A Cross-Sectional Study"

_diagnostics, 2023, doi:10.3390/diagnostics13101767_

Round 1

Reviewer 1 Report

The present paper introduced new method to identify of Jaundice ( high level of bilirubin) in adult patients using optical imaging and custom human brain like algorithm. Following are the comments:

·       This paper's introduction section is poorly written. It should clearly address the answers to the questions that follow.

·       What makes this proposed research novel? and why have authors addressed this issue? Are there any specific factors that helped to plan this proposed research?

·       The paper lacks recent relevant literature gaps. Authors should be required to explain why the proposed method is superior to the state-of-the-art techniques currently in use in terms of accuracy, robustness, etc.

·       The authors claimed to have taken photographs of patents' faces with their iPhone cameras. Are you doing this for a specific reason? Why had they not utilized a different tool for the same purpose?

·       How does the ABHB algorithm relate to the suggested research?

·       There's no need to start the introduction by talking about telemedicine. Please revise the entire introduction section so that it is relevant to the most recent jaundice start of art techniques.

·       The authors discussed study design and protocols in section 2.1, but there are no clinical protocols in this section. Please have a discussion about this.

·       Why did authors choose to use ABHB when there are so many other algorithms available?

·       Figures 3 should be redrew in high resolution with a large font size.

Also update the reference with recent publication and more related articles. 

·       Sketch2Photo: Synthesizing photo-realistic images from sketches via global contexts. Engineering Applications of Artificial Intelligence. doi: https://doi.org/10.1016/j.engappai.2022.105608

·       Automatic interpretation and clinical evaluation for fundus fluorescein angiography images of diabetic retinopathy patients by deep learning. British Journal of Ophthalmology. doi: 10.1136/bjo-2022-321472

·       New insights into natural products that target the gut microbiota: Effects on the prevention and treatment of colorectal cancer. Frontiers in pharmacology. https://doi.org/10.3389/fphar.2022.964793

·       Image Colorization using CycleGAN with semantic and spatial rationality. Multimedia Tools and Applications. doi: 10.1007/s11042-023-14675-9

·       Research on image classification method based on improved multi-scale relational network. PeerJ Computer Science. https://doi.org/10.7717/peerj-cs.613

·       Endoscope image mosaic based on pyramid ORB. Biomedical signal processing and control. doi: 10.1016/j.bspc.2021.103261

·       2D/3D Multimode Medical Image Registration Based on Normalized Cross-Correlation. Applied Sciences. doi: 10.3390/app12062828

After revision, this paper can be accepted.

Author Response

Point-by-point reply for Reviewer1

Q1: This paper's introduction section is poorly written. It should clearly address the answers to the questions that follow. What makes this proposed research novel? and why have authors addressed this issue? Are there any specific factors that helped to plan this proposed research?

The paper lacks recent relevant literature gaps. Authors should be required to explain why the proposed method is superior to the state-of-the-art techniques currently in use in terms of accuracy, robustness, etc.

Q3: How does the ABHB algorithm relate to the suggested research?

Q4: There's no need to start the introduction by talking about telemedicine. Please revise the entire introduction section so that it is relevant to the most recent jaundice start of art techniques.

Q6: Why did authors choose to use ABHB when there are so many other algorithms available?

A1,3,4,6: We appreciate your helpful suggestion. I thought your point was correct. We fully revised the introduction, discussing jaundice and recent technologies and eliminating the sentences about telemedicine. The meaning is easier to understand and better after the revision. We added the reason and advantage of ABHB: "Deep learning is commonly employed for image recognition and yields excellent accuracies [12,13]. However, a significant drawback of the technique is the requirement for extensive training data. To overcome such limitations, Zeta Bridge Corporation (Tokyo, Japan; https://www.zeta-bridge.com/) developed a human brain-like algorithm, recognizing that these image features can provide critical insights. By leveraging this approach, the algorithm can detect visual characteristics that traditional deep learning algorithms cannot, hence improving image recognition performance. The algorithm is named Algorithm Based on Human Brain (ABHB). ABHB was initially designed for deployment in foreign object detection systems on production lines. The ABHB's color analysis technology presents a novel approach to accurately obtain color information from images captured using ordinary cameras, regardless of the ambient light color conditions. Therefore, ABHB is independent of any light sources and does not require a color consistency patch. The present methodology enables an assessment to be conducted at any given time and location, obviating the necessity for specialized supplementary apparatus.”

Then we also fully revised the discussion section as “The study of objective color perception has a long history. In 1931, the Commission Internationale de l'Éclairage (CIE) proposed a color measurement system, which laid the foundation for contemporary colorimetry. This system enables the determination of color matches by employing CIE XYZ tristimulus values for color specification. The XYZ color space, also known as the CIE 1931 XYZ color space, is a mathematical representation of colors based on the trichromatic theory of human vision [18]. It exhibits "color differences."

The "BiliCam" study used the XYZ color space [9]. The authors dedicated their efforts to implementing ambient subtraction by capturing two images: one with flash and another without flash. On the other hand, Padidar et al. used HSI color space with machine learning [19]. They used a color calibration card and 100X microscope on the built-in camera of the phone. They reported that the estimation of bilirubin levels had a correlation of 0.479 with the total serum bilirubin values. Our result (0.528) was better than theirs without any color calibration cards, special microscopes, or datasets for machine learning.

Among color spaces, HSL, HSI, and HSV are perceptually based color models that simplify the representation of colors by separating hue, saturation, and lightness (or intensity/value) components. In graphic applications, HSL and HSI are two alternative color spaces used to represent colors. The conversion between XYZ and HSL/HSI enables the translation and manipulation of colors between these color spaces, catering to various color-related applications and requirements. HSL and HSI color spaces are generally regarded as equivalent. The main difference between HSL and HSI is in the way they define the brightness or lightness of a color. Notably, the HSL color space primarily concentrates on the perceptual dimensions of color, with a particular emphasis on the human perception of hue [20].

Most other studies using smartphones maintained consistent conditions by fixing the light source and distance or employing a color consistency patch [7 -11, 19, 21 - 24]. Typically, cameras modify color information via their built-in auto white balance function to minimize the effect of ambient light colors, ensuring that captured images appear natural when viewed by human observers. Due to variations in change logic across camera models and manufacturers, restoring accurate color information is currently a formidable task. measuring instruments of color images generally give different results depending on the light source conditions. The ABHB system presents an innovative approach that obviates the requirement for both, enabling any device to perform objective judgments akin to the cognitive processes of the human brain, regardless of the prevailing light source.

Deep learning exhibits great promise as a technology with substantial potential for further advancement within the realm of medical applications. Kalbande et al. evaluated two models, ResNet50 and Detectron-2, for detecting jaundice [13]. They employed a pre-trained dataset that underwent fine-tuning specifically for jaundice detection, resulting in an accuracy of approximately 95%. Although their results were commendable, the manuscript lacks pertinent details concerning the background data, including the levels of jaundice. Undoubtedly, deep learning will serve as an influential tool for image recognition. However, ABHB presents an alternative approach that circumvents the necessity for extensive datasets.

We have previously reported on predicting risky esophageal varices by only taking an abdominal picture using a smartphone application [25]. We believe that in the future, the acquisition of physical parameters from images of patients will become a standard practice in clinical settings, particularly in telemedicine and self-medication.” and added related references.

Q2: The authors claimed to have taken photographs of patients’ faces with their iPhone cameras. Are you doing this for a specific reason? Why had they not utilized a different tool for the same purpose?

A2: We added the sentence, “The rationale behind selecting a smartphone camera as the imaging device instead of a specialized camera was rooted in the fact that approximately 5 billion individuals currently utilize mobile devices [17]. Therefore, this study intended to leverage the popularity of smartphone cameras and utilize them as medical imaging devices.” on page 3, lines 109-113.

Q5: The authors discussed study design and protocols in section 2.1, but there are no clinical protocols in this section. Please have a discussion about this.

A5: We added the clinical protocols as “The inclusion criteria for this study were adult participants aged between 20 and 79 years, presenting with jaundice. The disease group comprises adults diagnosed with jaundice (total bilirubin≥ 3 mg/dl according to the classical definition [14], while the control group must be free of jaundice (total bilirubin < 3 mg/dl) upon blood testing. Participants exhibiting ocular conjunctival hyperemia or ophthalmologic conditions are excluded from the study. Additionally, the investigator reserves the right to exclude any participant deemed unsuitable for inclusion in the study.” on page 3, line 97-104.

Q7: Figures 3 should be redrew in high resolution with a large font size.

A7: We changed Figure 3 to a higher resolution with a large font size.

Q8: Also update the reference with recent publication and more related articles.

A8: Thank you for suggesting recent publications. We thoroughly read them and discussed and added to the reference; however, some do not seem related to our study. Therefore, we added the updated and related references, including your suggested article, as follows.

  1. Gao Z; Pan X; Shao J; Jiang X; Su Z; Jin K; Ye J. Automatic interpretation and clinical evaluation for fundus fluorescein angiography images of diabetic retinopathy patients by deep learning. Br J Ophthalmol. 2022, bjophthalmol-2022-321472. doi: 10.1136/bjo-2022-321472
  2. Kalbande, D., Majumdar, A., Dorik, P., Prajapati, P., Deshpande, S. Deep Learning Approach for Early Diagnosis of Jaundice. In International Conference on Innovative Computing and Communications. Lecture Notes in Networks and Systems; Gupta, D., Khanna, A., Hassanien, A.E., Anand, S., Jaiswal, A.; Springer, Singapore, 2023, Volume 492; pp. 387–395. doi: 10.1007/978-981-19-3679-1_30
  3. Aune A, Vartdal G, Jimenez Diaz G, Gierman LM, Bergseng H, Darj E. Iterative Development, Validation, and Certifi-cation of a Smartphone System to Assess Neonatal Jaundice: Development and Usability Study. JMIR Pediatr Parent. 2023, 28;6:e40463. doi: 10.2196/40463.

Reviewer 2 Report

This paper reported a novel method to assess potential jaundice during telemedicine using cell phone pictures. Overall the paper was well written and easy to understand. I feel it was a pleasure reading it. Plus, the topic is quite interesting and would bring great value to clinical medicine.

I only have 2 questions for the authors.

1. Please use "26" instead of "twenty-six" in abstract, line 23.

2. For the control group, what diseases were the "others"? Any disease could also cause jaundice, or digastric system related? It'll be great to add a brief explanation here.

Author Response

Point-by-point reply for Reviewer2

Thank you for your review.

Q1: Please use "26" instead of "twenty-six" in abstract, line 23.

A1: We changed it to 26.

Q2: For the control group, what diseases were the "others"? Any disease could also cause jaundice, or digastric system related? It'll be great to add a brief explanation here.

A2: We added the footnote for Table 1.